# Cardiovascular Morbidities in Adults Born Preterm: Getting to the Heart of the Matter!

**DOI:** 10.3390/children9121843

**Published:** 2022-11-28

**Authors:** Vasantha H. S. Kumar

**Affiliations:** Division of Neonatology, Department of Pediatrics, John R Oishei Children’s Hospital, University at Buffalo, 1001 5th Floor Main Street, Buffalo, NY 14203, USA; vkumar3@buffalo.edu; Tel.: +1-716-323-0260; Fax: +1-716-323-0294

**Keywords:** adults born preterm, hypertension, metabolic syndrome, heart failure, diabetes, ischemic heart disease, prematurity

## Abstract

Advances in perinatal and neonatal care have led to improved survival of preterm infants into adulthood. However, the shift in focus to long-term health in adults born preterm requires a clear understanding of the impact of prematurity on developing organ systems and the development of adult-oriented disease. A less well-recognized area of risk for surviving preterm infants is their cardiometabolic health. Epidemiologic evidence has linked preterm birth to the development of systemic hypertension, type 2 diabetes, metabolic syndrome, heart failure, and ischemic heart disease. Of more significant concern is that the risk of cardiometabolic disorders is higher in adults born preterm compared to full-term infants. The interconnected nature of the cardio-pulmonary system means worsening morbidity and mortality in adults born preterm. Addressing the problems of adults born preterm holistically would help promote cardiovascular health, wellness, and quality of life over their lifetime. Recognizing that adults born preterm are a unique subset of the population is a challenge in the current healthcare environment. Addressing issues relevant to adults born preterm in the clinically and research domain, using technology to characterize cardiopulmonary physiology and exercise tolerance, developing screening tools for early diagnosis and treatment, and robust follow-up of these infants with access to longitudinal data would improve both the quality and longevity of life in adults born preterm.

## 1. Introduction

Preterm birth is a leading cause of morbidity and mortality in children younger than five years of age [1]. Fortunately, advances in neonatal care, including antenatal corticosteroids, surfactant administration, gentle ventilation, parenteral nutrition, and provision of maternal breast milk have improved outcomes for infants born preterm. However, surviving infants are at greater risk of long-term morbidities such as bronchopulmonary dysplasia (BPD), cerebral palsy, feeding difficulties, and adverse neuro-developmental outcomes. Accompanying these morbidities are the high costs to the health care systems and financial and psychological hardships to families.

With increasing incidence and improving outcomes, prematurity has become a major global clinical and public health problem in most countries [2]. The majority of the preterm infants survive into adulthood and remain healthy and well [2,3]. Despite these improvements, premature infants have higher risks for morbidities and mortality than those born at term [4]. In a national cohort study, low gestational age at birth was independently associated with increased mortality in early childhood and young adulthood [5]. Lower gestational age was primarily associated with higher mortality in young adults from associated conditions such as congenital anomalies, endocrine, cardiovascular and respiratory disorders [5]. Furthermore, significant number of premature infants are at risk for cardiometabolic and cardio-pulmonary morbidities as adults [6], resulting in impaired exercise tolerance [7], pulmonary hypertension [8], and systemic hypertension [9] as adults. Preterm birth is increasingly recognized as a risk factor for cardiorespiratory disease [6] and an inverse association exists between birth weight and deaths from cardiovascular disease in adults [10,11]. The current review will focus on the nascent topic of cardiovascular problems of former premature infants and the challenges they face as they grow into adults.

Infants at term are born with adequate maturity of the cardiopulmonary system to sustain life as independent beings breathing room air at perinatal transition. However, premature birth or very low birth weight (VLBW) places these infants at risk for abnormal perinatal transition secondary to the immaturity of the cardiopulmonary system. Most infants born extremely preterm require respiratory support and oxygen therapy for prolong periods leading to development of bronchopulmonary dysplasia (BPD). With the advent of surfactant and gentle ventilation, ‘new’ BPD is predominantly associated with disruption of alveolar and microvascular development resulting in altered lung development in extremely premature infants. In this review, we will discuss the emerging evidence relating to the altered structure and function of the cardiovascular system as these infants survive into adulthood.

## 2. Cardiac Structure and Function in Adults Born Preterm

Alterations in cardiac structure and function during development results in substantially higher lifetime risks for cardiovascular disease in adults born preterm. Studies have utilized both echocardiography [12,13] and cardiac magnetic resonance imaging (MRI) [14,15,16] as tools in the assessment of cardiac structure and function. Extremely preterm infants exhibit a unique cardiac phenotype characterized by smaller left ventricles with altered systolic and diastolic function by early childhood [13]. Additionally, these infants had an increase in blood pressure with a decrease in left ventricular (LV), and aortic size with preserved LV function in adolescence [12]. In an interesting cross-sectional cohort study, Goss et al., evaluated premature infants born at 28 weeks gestational age (GA) at adolescence (13 years of age) and as young adults (25 years of age) by cardiac MRI [14]. Premature infants in both cohorts had a smaller biventricular cardiac chamber than the term group on MRI [14]. In addition to lower biventricular mass, the left-ventricular diastolic volume (LVDV) index and the left-ventricular systolic volume (LVSV) index were significantly smaller than in term infants in both the adolescent and the adult cohort [14]. Cardiac strain analysis demonstrated a hypercontractile heart, primarily of the right ventricle (RV), in adults born preterm [14]. Although premature infants were born in the broader time-period (the 1980s through 2000s), all infants demonstrated a lower cardiac mass, suggesting that prematurity may contribute to alterations in the heart’s structure in adults born preterm.

In former premature infants born at 30-week gestation, Lewandowski et al. demonstrated a greater RV mass, smaller RV volume with a significantly lower RV systolic functional parameters than term infants [12]. Furthermore, these premature infants had a higher LV mass along with a smaller LV with significant reductions in systolic and diastolic function parameters as adults [15]. Higher biventricular mass and hypo contractile strain pattern by Lewandowski et al. [15,16], are in striking contrast to the findings of reduced LV mass and hypercontractile strain pattern on MRI observed by Goss et al. [14]. It is interesting to note that the gestational age was comparable at birth between the two studies (around 28–29 weeks GA). However, the infants were born at differing time-periods ranging from 1980s to early 2000s (Lewandowski et al.: 1982–1985 [15,16]; Goss et al. 2003–2004 [14] (adolescents); 1980s–1990s: adult cohort). Likewise, the infants had cardiac MRI at differing ages as adults (Lewandowski et al.: 20–40 years; Goss et al.: 25 years). Cardiac development is a consequence of multiple prenatal, postnatal, and environmental factors. Both the studies are important and consistent in concluding that cardiac development is significantly altered in adults born preterm. Future studies addressing the factors that predispose to predominant involvement of either RV or LV, would help us in developing therapeutic strategies to optimize cardiac development in premature infants.

Extremely premature infants (<24 weeks GA at birth) are at high risk for cardiorespiratory morbidities as children and adults; however, there is paucity of data on the evolution of cardiac structure and function in extremely premature infants as they grow as adults. Extremely low birth weight (ELBW) infants are exposed to oxygen-rich environments soon after birth. Hyperoxia and generation of reactive oxygen species (ROS) result in the cell-cycle arrest of cardiomyocytes [17]. In contrast, ROS scavenging or inhibition of DNA damage response in postnatal hypoxia prolongs the postnatal proliferative window of cardiomyocytes [17]. Additionally, the proliferation of neonatal and embryonic stem (ES)-cell-derived cardiac cells involves ROS-mediated signaling cascades associated with NADPH oxidase in the cardiovascular differentiation of embryonic stem cells [18]. The interplay between the degree of cardiomyocyte immaturity, oxygen administrated, and ROS generation may play an essential role in both the differentiation and proliferation of cardiac myocytes soon after birth. These early structural and proliferation changes in the myocardium may significantly impact cardiac mass and function as premature infants grow into adults, predisposing them to long-term cardiac vulnerability [19]. In a rat model of chronic lung disease of prematurity aged one-year, adult rats demonstrated significant RV hypertrophy and dysfunction with significant chronic pulmonary hypertension following postnatal hyperoxia [20]. Hyperoxia-exposed RV cardiomyocytes showed evidence of mitochondrial dysregulation and DNA damage, suggesting potential mitochondrial dysfunction as a cause of RV dysfunction [20]. Studies should further examine the interactions between hyperoxia, BPD, and prematurity and how they impact the differentiation and proliferation of cardiomyocytes, leading to RV dysfunction and long-term cardio-respiratory morbidity in preterm adults.

Prematurity is associated with global alterations in myocardial structure and function, which worsen over time and potentially produce cardiovascular disease as these infants grow into adults. VLBW survivors who demonstrate reduced physical activity and impaired lung function, have altered left ventricular structure and function, as noted by reduced mass, size, stroke volume, and cardiac output [21]. Beyond the effects of physical activity and body mass index, lung function and cardiac structure and function contributed equally to reduced exercise capacity [21]. Cardiac structural alterations have been noted as early as six years of age in ELBW infants [13]. The LV and the RV are equally affected by structural alterations, as demonstrated by cardiac resonance imaging [15,16]. Longitudinal studies suggest that alterations in cardiac structure precede alterations in cardiac function in adults born preterm. The pathophysiology of the cardiovascular morbidities in adults born preterm is illustrated in Figure 1.

## 3. Risk for Cardiovascular Disease

Studies have shown that endothelial dysfunction, an established early marker of the development of hypertension and cardiovascular disease, may occur earlier in premature infants. Compared to term controls, reduced endothelial function has been noted in premature-born adults as measured by finger plethysmography [19] or flow-mediated dilatation [20]. However, the results must be interpreted with caution as factors such as dyslipidemia, impaired glucose responses, circulating sex hormone levels, and undetected atherosclerotic vascular disease alter endothelial function. Nevertheless, the evidence points to an increasing group of adults born preterm with risks of long-term complications and early adult death [22]. The adjusted hazard ratios (aHRs) for all-cause mortality were 1.44 (95% CI, 1.34–1.55) for moderate preterm birth (23 to 33 weeks gestation); 1.23 (95% CI, 1.18–1.29) for late preterm birth (34 to 36 weeks); and 1.12 (95% CI, 1.09–1.15) for early term deliveries (37 to 38 weeks). Preterm birth is associated with two-fold increased risks of death from cardiovascular disease (aHR, 1.89; 95% CI, 1.45–2.47) and diabetes (aHR, 1.98; 95% CI, 1.44–2.73) [22]. Accordingly, it is imperative to detect modifiable risk factors or early signs of cardiovascular disease in premature infants, to reduce the lifelong disease burden on individuals and society.

### 3.1. Diabetes and Cardiovascular Risk

In adult life, the risks for type 2 diabetes are higher in premature infants born less than 35 weeks of gestation and the risk is independent of fetal growth [23]. Additionally, VLBW infants had diminished sensitivity to insulin compared to term infants with comparable body size and composition as adults [24]. Nonetheless, higher insulin secretion compensates for depressed sensitivity to insulin in adults born preterm. The emphasis on healthy lifestyles and expeditious screening for type 2 diabetes is crucial as these infants may have predisposition to glucose intolerance as adults [24]. The metabolic profile of VLBW infants can be difficult to ascertain as higher insulin resistance is not necessarily accompanied by dysglycemia as adults [25]. In population-based studies, a 2-fold increase in the prevalence of type 2 diabetes has been noted by 40 to 60 years of age in infants born preterm [23,26]. In a recent meta-analysis, infants born preterm had a higher risk of diabetes and hypertension as adults [27]. Early diagnosis and management of prediabetes would contribute not only to decrease the disease progression towards type 2 diabetes [28] but also in reducing cardiovascular morbidity in these infants [29].

### 3.2. Metabolic Syndrome and Cardiovascular Risk

Body composition studies in former preterm infants have shown a lower lean body mass, an increase in body fat, and a higher risk of developing dysglycemia by the fourth decade of life [30]. Similar results relating to parameters of body composition have been reported in other studies in 20-year-old adults born preterm [31]. As body composition and metabolic disturbance may reveal with advancing age, long-term follow-up is essential for early diagnosis and appropriate intervention in these adults at increased risk for cardiovascular disease. Development of adipose tissue essentially occurs in the third trimester of pregnancy. Disturbances in adipocyte development during the third trimester could have consequences on adipocyte function throughout the life course in adults born preterm [32]. As obesity and body fat are major predictors of dysglycemia, it is essential to evaluate these infants and develop lifestyle interventions and wellness behaviors to modify body fat composition of the preterm population in young adulthood. However, there is paucity of data regarding when the lifestyle interventions should begin for maximum benefit in these infants. Nonetheless, it would be advantageous to start these interventions as early as possible to minimize the effects of adult-onset disease [30]. Prenatal and postnatal factors around the time of birth determine body composition relating to adiposity development and deposition [33]. The study highlights the emphasis on lifestyle interventions and wellness habits during the critical period of development to change modifiable factors during early childhood for potential benefits on cardiometabolic risk in later life.

Metabolic syndrome is a cluster of risk factors such as high blood pressure, excess body fat, high blood sugar and abnormal cholesterol levels that increase the risk for cardiovascular disease and stroke. Co-existence of more than one factor results in evolution of systemic inflammation [34] or ongoing oxidative stress [35], further increasing the risk for cardiovascular disease. A marked increase in risks factors such as hypertension, airflow obstruction and glucose intolerance has been noted in young adults born preterm [36]. However, the presence of these risk factors did not necessarily increase inflammatory or oxidative stress markers in these infants compared to full term infants [36], implying different pathogenetic mechanisms. The pathophysiology of the development of these risk factors may be rooted in interactions between developing organ systems over time. For instance, smaller kidneys noted in premature infants along with abnormal urine albumin to creatinine ratio and elevated angiopoietin-1 levels may suggest a renal etiology for elevated blood pressure [37]. An increase in the levels of antiangiogenic proteins such as soluble fms-like tyrosine kinase-1 may suggest microvascular hypoplasia and capillary rarefaction resulting in higher blood pressure [38]. Multiple developmental mechanisms impact cardiovascular development, further modified by postnatal factors and lifestyle changes as premature infants grow into adults. Epigenetic programming and neonatal conditions further complicate the development of cardiovascular risk factors leading to adult-onset diseases.

Studies have tried to quantify cardiometabolic risk using variables such as central obesity or waist circumference, blood glucose level, systolic blood pressure, high density lipoproteins and triglyceride levels [39]. In a study that included the above variables, infants born moderately preterm, had higher mean cardiometabolic risk scores at age 3 to 12 years compared to full-term infants [40]. Interestingly, each additional week of gestational age was associated with lower cardiometabolic risk score in premature infants [40]. As cardiometabolic risk can be tracked throughout childhood into adulthood, lifestyle and behavioral interventions in early childhood may prevent the development of metabolic syndrome and morbidities from cardiovascular disease in adults born preterm [40].

## 4. Adult-Onset Hypertension

Epidemiologic evidence has linked preterm birth with risks of hypertension [41] and type 2 diabetes [42], with smaller and less mature infants facing the most significant risks as young adults. In a systematic review, preterm infants with a mean GA of 30 weeks at birth had higher systolic blood pressures by 3.8 mmHg compared to term infants at 18 years of age [41]. Similarly, VLBW infants had higher systolic and diastolic blood pressures as adults, with female gender and preeclampsia being additional risk factors [9]. Young adults born <28 weeks gestation in the post-surfactant era had higher 24 h systolic (Mean: 4.5 mHg), diastolic (Mean: 3.4 mmHg) and mean (Mean: 3.6 mmHg) BP and an increase in ambulatory BP compared to term controls [43]. Additionally, the visit-to-visit variability to assess BP patterns may help identify young adults at increased risk of cardiovascular disease and all-cause mortality later in life [44]. Furthermore, the potential role of the so called “multiple office blood pressure measurement” still need to be studied [45]. Both healthy and sick preterm infants are at increased risk for cardiovascular morbidities such as high blood pressure in life [46]. In a systematic review of >17,000 preterm infants born < 37 gestational weeks adults born preterm had a significantly higher SBP (mean of 4.2 mm Hg), DBP (mean of 2.6 mm Hg), and low-density lipoproteins (LDL) as compared to adults born at term [47]. Women had relatively higher differences in blood pressure than men, suggesting that sex is an essential variable in preterm-born adults [47]. However, both men and women are at risk for adult-onset hypertension, and studies should enlighten whether sex is an essential variable for adult-onset hypertension in preterm adults. Interestingly, a recent study found that pulse-wave velocity, with advancing age, may behave differentially in women as compared to men [48].

Adult-onset hypertension can partly be explained by structural changes in the vessels in adults born preterm. Premature infants born around 30 weeks of gestation had 20% smaller thoracic and abdominal aortic lumens but similar carotid and brachial diameters as adults at 23 to 28 years of age by magnetic resonance [49]. Despite similar carotid size, the pulse wave velocity was increased, and carotid distensibility decreased in adults born preterm [49]. Variations in blood pressure and early aortic elastin and collagen developmental changes in the aorta in infants born preterm could explain some of the changes seen as adults. These changes could increase over the lifetime and may increase the risk of vascular aging in adults born preterm [49]. Higher serum creatinine and NGAL in the preterm group may indicate that preterm birth may affect kidney function and may partly explain more elevated blood pressure in preterm-born adults [50]. Close monitoring of blood pressure over time to facilitate early detection and timely management of hypertension is of utmost importance to improve the quality of life in this high-risk population.

## 5. Preterm Birth and Ischemic Heart Disease

Preterm birth (<37 weeks GA) has been linked with an increased risk of cardiometabolic disorders in adulthood, including hypertension [51], diabetes [52], and metabolic syndrome [24,25], which are the significant risk factors for ischemic heart disease. In a population-based cohort study from Sweden (age range of 18 to 43 years), gestational age was inversely correlated with the risks for ischemic heart disease (adjusted HR per additional week of gestation of 0.96; 95%CI, 0.93–0.98) [53]. The adjusted odds of ischemic heart disease for those born preterm (<37 gestational weeks) and those born early term (37–38 gestational weeks) were 1.44 (95% CI, 1.19–1.73) and 1.16 (95%CI, 1.02–1.31), respectively, compared to term infants. It is interesting to note that the risks for ischemic heart disease were correlated with gestational age in late adulthood (30 to 43 years) but not in early adulthood (18 to 29 years) [53]. The risks for ischemic heart disease in late adulthood were higher in preterm and early term infants compared to term infants (infants <37 weeks—adjusted HR: 1.53; 95% CI, 1.20–1.94; early term infants—adjusted HR: 1.19; 95% CI, 1.01–1.40, respectively). Of note, the increased risk for ischemic heart disease was mostly driven by infants born between 34- and 36-weeks’ gestation, due to limited number of babies < 34 weeks gestational age enrolled in the study [53]. It is interesting to note that on sex-stratified analysis, preterm birth was associated with significantly increased relative risk among women but not in men [53]. In one of the oldest longitudinal studies, with the birth cohort between 1924 and 1944, premature infants (<34 weeks versus 34–37 weeks at birth) were followed up to old age and found no difference in coronary heart disease in infants born preterm [54]. Differing results from the above two studies may be related to nutritional and environmental factors along with generational differences in the study population, as well as to the fact that in 1924–1944, preterm infants were likely to die of other diseases and complications before reaching an age where cardiovascular disease would have impacted life expectancy. Improved medical care and increased neonatal and pediatric survival might have relevantly shaped this result.

Increased risk of ischemic heart disease is reported among adults with either average or low fetal growth. Intrauterine environment contributing to low fetal growth may influence the earlier development of cardiovascular risk factors leading to higher mortality in adults [55]. Multiple mechanisms in the perinatal period increase the risk for ischemic heart disease in adults. Firstly, the association of preterm birth with hypertension, diabetes, metabolic and placental dysfunction, and insulin resistance potentially increases the risk of developing ischemic heart disease in adults [56,57]. Secondly, by disrupting intrauterine growth, preterm birth could compromise cardiovascular structure and function. Finally, the association of preterm birth with the higher incidence of lipid disorders in early to mid-adulthood potentially worsens the development of ischemic heart disease [58]. Persons born preterm may need early preventive evaluation and long-term monitoring for lipid profiles and the development of metabolic syndrome to reduce their future cardiovascular risks. Longitudinal studies addressing the long-term cardiovascular risks and outcomes in extremely low birth weight infants <32-week gestation would further help in understanding perinatal origins of adult disease.

## 6. Preterm Birth and Reduced Exercise Capacity

In a cohort of young adults born small for gestational age (SGA), exercise capacity was markedly reduced with decreased maximal workload [59]. In a population-based study, very low birth weight (VLBW) infants demonstrated reduced oxygen uptake, work rate, and oxygen pulse at peak exercise and earlier anaerobic thresholds at 26–30 years of age [21]. Alterations in cardiac structure and function, specifically of the left ventricle along with impairment in lung function may lower the threshold for exercise capacity in these infants. Further research into reduced exercise capacity in SGA infants is warranted to uncover the potential association between increased cardiovascular mortality and reduced exercise capacity among adults born SGA.

## 7. Heart Failure in Adults Born Preterm

Unlike neurodevelopmental issues, which often lessen over time, cardiometabolic risks are invisible in childhood, only to emerge in adolescence, young adulthood, or even middle age. In a population-wide study, preterm birth <37 weeks GA was associated with an increased incidence of heart failure in infancy, childhood, and adulthood [60]. Adult-onset heart failure was 4.7-fold higher for preterm infants compared to term controls; the risks in late preterm infants were 1.2-fold higher than in full-term infants [60]. Heart failure may be partially mediated by hypertension, diabetes, dyslipidemia, ischemic heart disease, and sleep-disordered breathing [51,53,58,61,62], which are known to be associated with preterm birth. In addition, cardiac imaging studies in young adults have found evidence of cardiac remodeling in those born preterm, including LV mass, reduced diastolic relaxation, RV dysfunction, and reduced myocardial reserve during physiologic stress [14,15,19].

Severe BPD, a complication of extreme immaturity of the lungs, frequently leads to pulmonary hypertension and may contribute to right-sided heart failure. Additionally, traditional risk factors such as hypertension and diabetes are higher in preterm infants. Alterations in the structure and function of the myocardium may play an essential role in the development of heart failure in these infants. At least in animal models, rats exposed to postnatal hyperoxia had significantly greater RV hypertrophy, RV systolic pressures, reduced RV ejection fraction, and significant RV-PA uncoupling [20]. Hyperoxia-exposed RV cardiomyocytes demonstrated mitochondrial dysregulation and mitochondrial DNA damage, suggesting mitochondrial dysfunction as a cause of RV dysfunction in adult rats [20]. Impaired RV-PA coupling from high resistance-low compliance pulmonary vascular beds with attenuated RV adaptation in the face of increased vascular load has been demonstrated in young adults born preterm [63].

Furthermore, young male adults born preterm with no overt cardiopulmonary disease exhibit altered pulmonary microvascular hemodynamics as measured by magnetic resonance imaging, suggesting that prematurity may have persistent lifelong consequences on pulmonary vascular health [64]. The extent of PA-RV coupling is a critical determinant of overt clinical symptoms of right heart failure such as peripheral edema, hepatomegaly, palpitations and reduced exercise tolerance. Female patients with pulmonary arterial hypertension (PAH) adapt better to an increased afterload by increasing RV contractility than male patients [65]. The role of sex hormones in developing RV dysfunction and RV failure could have implications in adults born preterm [66].

## 8. Assessing Cardiovascular Health in Adults Born Preterm

Innate ‘programming’ of cells and tissues can be altered by prematurity from changes in metabolic pathways and signaling molecules resulting in broad changes in the structure and function of organ systems as these infants grow into adults. Factors such as fetal growth restriction, maternal preeclampsia, chorioamnionitis, and placental dysfunction can disrupt cellular mechanisms, producing phenotypic changes in organ systems in premature infants. The fetal origin of adult disease has been eloquently described by Barker et al. [55]. The potential for changes in the vasculature, including elastin and collagen deposition, could lead to the development of aortic stiffness over time, contributing to increased morbidity and vascular aging. Assessing preterm-born adults for adverse effects from changes in organ systems is a massive undertaking and requires a multidisciplinary approach to manage these patients. Some of the significant cardiovascular morbidities in adults born preterm are illustrated in Figure 2.

National organizations have not established any guidelines in the evaluation of adults born preterm. However, increased survival of premature adults without significant comorbidities means they need lifetime follow-up. Assessment of premature infants once discharged from the NICU is primarily the responsibility of the Pediatrician, who in general is more geared towards wellness in the general population. However, as they grow into adults, their transition to adult Physicians such as family practice or internal medicine further dilutes the follow-up of these patients. Critical pieces of historical information of the patient such as the degree of prematurity, birth weight, mechanical ventilation, and significant or minor comorbidities should be obtained by all Physicians. Physicians should be proactive in both assessment and management of adults born preterm, to promote wellness and lifestyle changes to reduce cardiometabolic risks, hypertension, and ischemic heart disease in the population.

Despite the large amount of resources invested in the survival and care of extremely premature infants, it is surprising that minimal resources are invested in their long-term follow-up. A commitment to wellness involves a regular, extensive follow-up of adults born preterm. It may include regular monitoring of blood pressure, annual assessment of cardiometabolic risk profile, and evaluating cardiac performance and cardiovascular fitness by echocardiography and cardiopulmonary exercise testing. These are not clinical guidelines. However, future studies should address optimal approaches in the evaluation and management of adults born preterm. A proactive approach and early diagnosis should be the guiding principle for disease management in this high-risk population. Exercise tolerance monitoring can be used as a screening tool to assess the integrity of the cardio-respiratory system. More recently, a novel home-based exercise intervention that avoids multiple hospital visits and uses technology to comprehensively characterize cardiopulmonary physiology in children and adolescents born extremely preterm has been proposed [67]. Adults born preterm may have a combination of abnormalities that could impact lung and cardiac structure and function, which may be challenging to tease out clinically. The risk for multiple complications and cardio-respiratory issues, including endocrine, neurological, and mental health problems, could burden the adults born preterm. Addressing the issues in a multidisciplinary approach would help promote cardiovascular health, wellness, and quality of life over the lifetime (Figure 3).

## 9. Future Research Directions

Advances in neonatal care have contributed to increased survival and a reduction in major morbidities in premature infants. Additionally, nutritional best practices have resulted not only in the appropriate growth but also have reduced the length of stay of premature infants in the intensive care unit. I am personally optimistic that these advances would have long-term benefits as premature infants grow as adults. However, cardiometabolic risks may be inherent in adults born preterm from prematurity itself.

The relative contribution of fetal versus postnatal changes in tissue and organ development in the context of extremely preterm infants is essential to understand the long-term morbidities in adults born preterm. Studies should address the development of altered structure and function of the myocardium over time across extremes of gestational age. Specific information on the role of in utero (preeclampsia or placental dysfunction) or postnatal determinants of right versus left heart failure, systolic versus diastolic dysfunction, and the evolution of myocardial changes will help identify distinct causes and increase the potential for therapeutic interventions. Is there a way to ‘predict’ heart failure depending on cardiometabolic risk factors? Would screen for early, asymptomatic signs of altered cardiac structure on imaging, electrocardiography or by using biochemical markers at discharge or in early childhood predict the development of clinically significant heart failure over time? Is there a role for Brain Natriuretic Peptide (BNP) as a biomarker in the early prediction of heart failure in these infants? Pulmonary hypertension has been studied in depth; however, there is a lack of data on the impact of the severity of pulmonary hypertension and its interaction with myocardial structural changes seen in these infants in the development of heart failure over time. The story of adult-onset systemic hypertension and its relationship to vascular changes over time in adults born preterm would be of interest. The relationships between fetal growth, prematurity, and postnatal nutrition on arterial health and its impact on the development of hypertension in adults born preterm need further exploration. Interactions between the cardiovascular and respiratory systems in the development of exercise intolerance or cardio-respiratory compromise need further study. A higher risk of metabolic syndrome, diabetes, and ischemic heart disease in adults born preterm should be highlighted by the American Medical Association and the American Heart Association. National organizations’ commitment to develop guidelines and screening tools would help in appropriate management and further research into cardiovascular issues in adults born preterm.

It is known that infants born small for gestational age develop exercise intolerance and may be associated with higher cardiovascular mortality as adults. Understanding the causative relationships between impaired cardiopulmonary function and reduced aerobic exercise capacity in ex-preterm adults is essential. One area that needs focus is the role of nutrition and optimal growth in infants born prematurely, which may represent modifiable factors in shaping cardiovascular risk factors as they grow into adults. More research, particularly on the role of breast milk in enhancing cardiac performance and exercise capacity, would help understand early life interventions’ role in developing cardiac protective strategies in adults born preterm [68]. Are there factors modifiable while the infant is in the NICU, early in postnatal life, or even in early childhood that would improve outcomes in adults born preterm? Additional data in older adults born preterm should address whether they have a ‘prematurely aged’ cardiopulmonary system [69]. Research should focus on early and later life interventions designed to improve cardiac and lung function, increase exercise capacity, and reduce mortality in these infants.

## 10. Conclusions

The 2030 agenda for sustainable development adopted by all United Nations Member States in 2015 provides a blueprint for peace and prosperity for people and the planet, now and into the future. At its heart are the seventeen Sustainable Development Goals (SDGs), which are an urgent call for action by all developed and developing countries in a global partnership. One of the main goals is to ensure healthy lifes and promote well-being for all ages. With the increase in the incidence of infants born preterm, the problems of adults born preterm would increase exponentially over time in all countries. The burden of caring for and maintaining the wellness of these adults will be high. Developing strategies not only to prevent prematurity but also to maintain the quality of life would go a long way in contributing to wellness in adults born preterm.

Interactions between Pediatricians and adult Physicians at transition (18 to 25 years) are critical in maintaining wellness in these adults. The ‘Preterm Card’ would help adult physicians with the essential developments that occur in the first 18 to 25 years of life to facilitate care in the transition phase. It would also be the foundation on which adult physicians can develop diagnostics, strategies, and interventions to promote lifelong wellness in adults born preterm. In addition, research into the long-term implications of being born early in all organ systems will help us understand the complexities of the interaction among organ systems in maintaining wellness and aging over a lifetime. Getting to the ‘heart’ of the matter would contribute to cardiovascular health and enhance the quality of life in survivors of premature birth.

The development of academic programs specifically addressing the problems of adults born preterm and commitment of resources by all stakeholders would boost the ‘wellnesses’ of the preterm-born adults. The development of databases with all essential pieces of information with easy accessibility to physicians would help develop intervention strategies to improve the quality of life of adults born preterm. Addressing the development of cardiovascular risk factors and cardiac-related conditions and their interactions with other organ systems will contribute to the overall wellness of the adults born preterm.

## Figures and Tables

**Figure 1 children-09-01843-f001:**
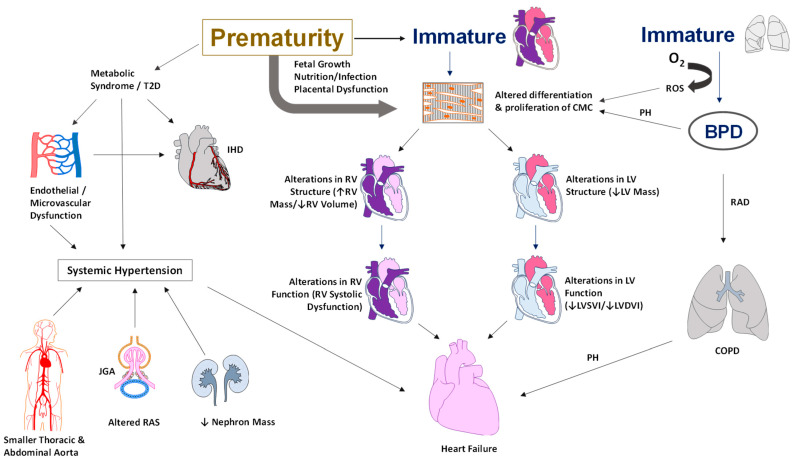
Illustration of the pathophysiology of cardiovascular morbidities in adults born preterm. Alterations in the cardiac structure include a smaller RV or LV systolic or diastolic volume and an increase in RV or LV mass. Functional changes include systolic and diastolic dysfunction, hypercontractile heart, and low cardiac output. Metabolic syndrome and a higher risk of type II diabetes contribute to endothelial dysfunction leading to systemic hypertension and ischemic heart disease. In addition, the smaller diameter of the thoracic and abdominal aorta, lower nephron mass, and alterations in sodium balance with activation of the renin-angiotensin system all contribute to the development of systemic hypertension in adults born preterm. Abbreviations: BPD—bronchopulmonary dysplasia, RAD—reactive airway disease, COPD—chronic obstructive airway disease, O2—oxygen, CMC—cardiomyocyte, RV—right ventricle, LV—left ventricle, T2D—type II diabetes, JGA—juxtaglomerular apparatus, RAS—renin-angiotensin system, PH—pulmonary hypertension, ROS—reactive oxygen species, IHD—ischemic heart disease, LVSVI—Left Ventricular Systolic Volume Index, LVDVI—Left Ventricular Diastolic Volume Index (copyright: Vasantha Kumar. H.S).

**Figure 2 children-09-01843-f002:**
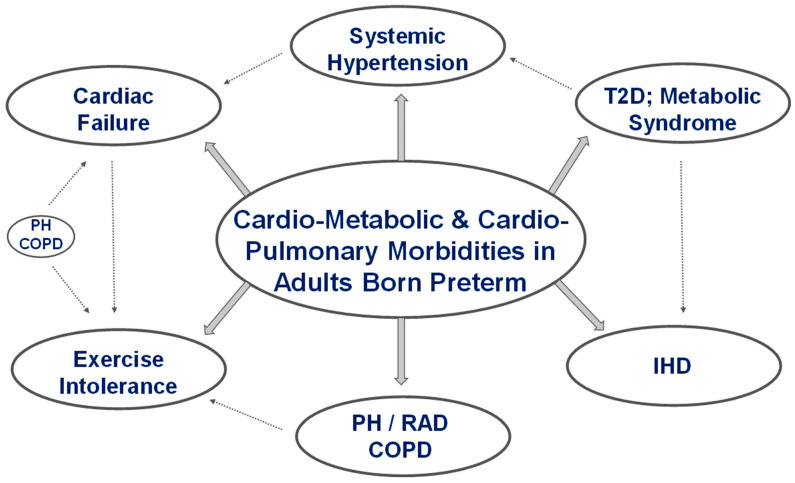
Diaphragmatic representation of clinical morbidities seen following premature birth in adults born preterm. The interconnectedness of the cardiac and pulmonary systems means that the heart carries the brunt of the effects of pulmonary morbidity. Early onset of type 2 diabetes and metabolic syndrome are risk factors for ischemic heart disease and systemic hypertension in adulthood. Systemic and pulmonary hypertension and chronic lung disease (COPD) are risk factors for the development of cardiac failure. Pulmonary hypertension, reactive airway disease and COPD could trigger exercise intolerance in preterm-born adults. Abbreviations: RAD—reactive airway disease, COPD—chronic obstructive airway disease, T2D—type II diabetes, PH—pulmonary hypertension, IHD—ischemic heart disease. (copyright: Vasantha Kumar. H.S).

**Figure 3 children-09-01843-f003:**
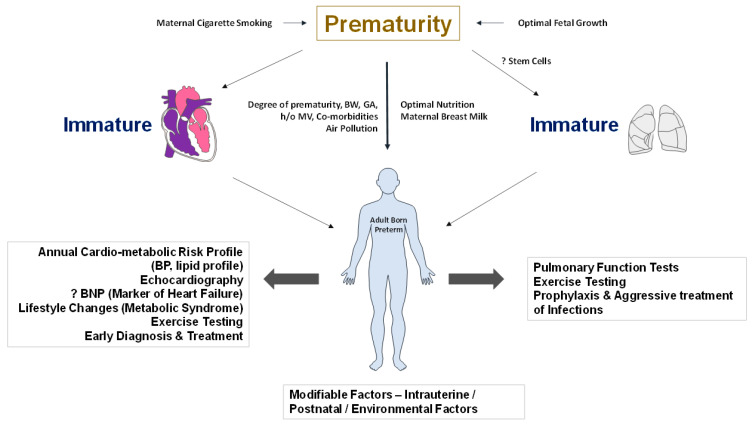
Screening, early diagnosis, and interventions that would improve the quality of life in adults born preterm. Maternal cigarette smoking and intrauterine growth retardation contribute adversely to fetal development. Optimal postnatal nutrition and maternal breast milk promotion may facilitate organ system growth patterns. Screening for metabolic syndrome, biomarkers for detection of heart failure, and pulmonary function tests for obstructive airway disease may help in the early diagnosis and management of comorbidities. Lifestyle changes to promote wellness are of particular importance in adults born preterm. The role of stem cells in promoting lung growth needs more studies. Abbreviations: BW—birth weight, GA—gestational age, MV—mechanical ventilation, BP—blood pressure, BNP—brain natriuretic peptide. (copyright: Vasantha Kumar. H.S).

## Data Availability

Not applicable.

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
