# Peer review of "Cardiovascular Morbidities in Adults Born Preterm: Getting to the Heart of the Matter!"

_children, 2022, doi:10.3390/children9121843_

Round 1

Reviewer 1 Report

This is a very thorough and relevant manuscript by Dr. Kumar.

Cardiovascular risk factors in adults born preterm are increasingly being recognized as we are seeing more of these patients survive. The exact mechanisms for such are not well known and the author does a good job by summarizing the current literature in a thoughtful manner.

A few comments:

-       Overall, the manuscript is well written

-       The manuscript could benefit from being concised and coherent. There is a lot of repetition in some sections that could be better organized. A few examples:

o   Lines 120-122 - sentence is long and could be reworded

o   In the adult-onset hypertension section - similar points could be summarized together - like mean difference in BP in adults born preterm; one study is quoted in line 276 and the other in 295. 

o   Lines 327-332 the discussion of exercise capacity seems out of place with the context of discussion in that section

o   Future directions section needs revision – it is too long in its current form with again a lot of repetition

Author Response

This is a very thorough and relevant manuscript by Dr. Kumar.

Cardiovascular risk factors in adults born preterm are increasingly being recognized as we are seeing more of these patients survive. The exact mechanisms for such are not well known and the author does a good job by summarizing the current literature in a thoughtful manner.

A few comments:

-       Overall, the manuscript is well written

-       The manuscript could benefit from being concise and coherent. There is a lot of repetition in some sections that could be better organized. A few examples:

 We thank the reviewer for the comments. The whole manuscript is thoroughly reviewed, and duplications and repetitions are removed as much as possible. The manuscript has been shortened by about 300 words.

  • Lines 120-122 - sentence is long and could be reworded
  • The author has corrected Lines 120-122.

  • In the adult-onset hypertension section - similar points could be summarized together - like mean difference in BP in adults born preterm; one study is quoted in line 276 and the other in 295. 

  • We thank the reviewer for noticing it. We have grouped all sentences relating to BP measurements. The author has reworked all sections relating to adult-onset hypertension.

  •  
  • Lines 327-332 the discussion of exercise capacity seems out of place with the context of discussion in that section
  • The author thanks the reviewer for the comment. The author has grouped all related information on exercise capacity in a separate subheading.

  • Future directions section needs revision – it is too long in its current form with again a lot of repetition
  • The author has reworked the section. Deleted information that is either not relevant or duplicated sentences. The section is significantly shortened.

Reviewer 2 Report

Thanks for the chance to review this manuscript. Please see below some comments:

Major: similarity report was very high. I suggest to re-check the whole manuscript.

Minor: 

1- Line 93 to 108 describe the assessment of heart structure and function in preterm infants by MRI from different studies. It would be great if the author add a table as well to present these studies in more graphical way.

2- It would be interesting to see author's opinion about premature infant born for COVID-19 infected mothers. I think it worth adding  a paragraph about this.

Author Response

Thanks for the chance to review this manuscript. Please see below some comments:

Major: similarity report was very high. I suggest to re-check the whole manuscript.

The author thanks the reviewer for comments and suggestions. The whole manuscript is thoroughly reviewed. Sentences that are duplicated and unrelated information has been removed. Most of the similarity index report is <1% for references and limited to words and not sentence structure. The author is confident that has been reduced. Being a review of literature, it is unavoidable at times for similarity to exist.

Minor: 

  • Line 93 to 108 describe the assessment of heart structure and function in preterm infants by MRI from different studies. It would be great if the author add a table as well to present these studies in more graphical way.

We have accomplished reviewer’s suggestion as much as possible in Figure.1.

2- It would be interesting to see author's opinion about premature infant born for COVID-19 infected mothers. I think it worth adding a paragraph about this.

As much as the author would like to add information on COVID-19, the review is about cardiovascular morbidity in adults born preterm. Although literature exists in pediatrics and in infants with congenital heart disease, there is paucity of literature on covid-19 and its long-term effects specifically in premature infants.  The author would keep reviewer’s suggestion in mind for the future.  

Reviewer 3 Report

I think is an interesting topic, however, the paper is not well written. The abstract is too extensive. In the introduction section, there are many paragraphs which have not references. Methodology and discussion are missing. What is the aim of the study?  Conclusions is not related with the text. 

Respect to figures, I don't know if the figures are yours or not, if not, where is the reference?

The paper should be rewritten and structured. Thank you for sharing your research. 

Author Response

I think is an interesting topic, however, the paper is not well written. The abstract is too extensive. In the introduction section, there are many paragraphs which have not references. Methodology and discussion are missing. What is the aim of the study?  Conclusions is not related with the text. 

The author thanks the reviewer for the suggestions. However, the author would like to add that the article is a review of literature on cardiovascular morbidities in adults born as premature infants. Being not a original study, we do not have methods, results and discussions. The author feel that the abstract is appropriate with 219 words. Most of the reviews and studies limit it to 250 words. The author has added additional references to the introduction section.

Respect to figures, I don't know if the figures are yours or not, if not, where is the reference?

The figures are authors own and hence copyrighted.

The paper should be rewritten and structured. Thank you for sharing your research. 

The author has extensively revised the manuscript. The manuscript is shorter by about 500 words. We have deleted unrelated information and duplicated sentences.

Round 2

Reviewer 3 Report

Thank you for your manuscript. I think it has been improved. 

Author Response

The reviewer has no comments.